# Potential and Possible Therapeutic Effects of Melatonin on SARS-CoV-2 Infection

**DOI:** 10.3390/antiox11010140

**Published:** 2022-01-09

**Authors:** Evgeny Shchetinin, Vladimir Baturin, Eduard Arushanyan, Albert Bolatchiev, Dmitriy Bobryshev

**Affiliations:** 1Department of Pathophysiology, Stavropol State Medical University, 355000 Stavropol, Russia; ev.cliph@rambler.ru; 2Department of Clinical Pharmacology, Stavropol State Medical University, 355000 Stavropol, Russia; prof.baturin@gmail.com; 3Department of Pharmacology, Stavropol State Medical University, 355000 Stavropol, Russia; prof.arushanyan@gmail.com; 4Center of Personalized Medicine, Stavropol State Medical University, 355000 Stavropol, Russia; bobryshevrg@yandex.ru

**Keywords:** melatonin, SARS-CoV-2, COVID-19, inflammation, cytokine storm

## Abstract

The absence of effective drugs for COVID-19 prevention and treatment requires the search for new candidates among approved medicines. Fundamental studies and clinical observations allow us to approach an understanding of the mechanisms of damage and protection from exposure to SARS-CoV-2, to identify possible points of application for pharmacological interventions. In this review we presented studies on the anti-inflammatory, antioxidant, and immunotropic properties of melatonin. We have attempted to present scientifically proven mechanisms of action for the potential therapeutic use of melatonin during SARS-CoV-2 infection. A wide range of pharmacological properties allows its inclusion as an effective addition to the methods of prevention and treatment of COVID-19.

## 1. Introduction

On 11 March 2020, the World Health Organization declared coronavirus disease 2019 (COVID-19) caused by SARS-CoV-2 a global pandemic [1]. SARS-CoV-2 is the seventh coronavirus that infects humans—SARS-CoV-2, SARS-CoV and MERS-CoV can cause a severe course of disease with fatal outcomes, while others: HKU1, NL63, OC43, and 229E cause mild disease [2,3]. According to comparative analysis of genomic data it has been clearly shown that SARS-CoV-2 is not a laboratory construct or a purposefully manipulated virus [3]. Therefore, we can assume that in the future there may be other pandemics caused by new viruses, including new coronaviruses, or new variants of SARS-CoV-2. For example, a new variant of the virus, Omicron, was recently identified in South Africa. Preliminary data showed that the neutralization efficiency of Omicron of those vaccinated with BNT162b2 has decreased [4]. Thus, it is obvious that it is necessary to constantly search for new methods of treatment and prevention of viral diseases [5,6]. 

More than 60% of COVID-19 cases are asymptomatic or mild [7]. According to the United States Centers for Disease Control and Prevention (CDC) and the Chinese CDC, severe disease with hospitalization (e.g., with dyspnea, hypoxia, or >50% lung involvement on imaging within 24 to 48 h) was reported in 14%; critical disease (e.g., with respiratory failure, shock, or multiorgan dysfunction) was reported in 2–5%; the overall case fatality rate was 2.3–5% [8,9]. So far, the basic preventive measures of infection control, including social distancing, hand washing and wearing a mask, and vaccination have been shown to be effective with a high degree of evidence [10,11]. There is evidence of the effectiveness of some therapeutic interventions for severe cases of infection [12,13], but despite this, mortality from COVID-19 remains at a high level [14,15,16]. 

Currently, there are no specific, prognostically effective, and safe treatments for mild to moderate COVID-19, reducing the likelihood of disease progression, length of hospital stays, need for mechanical ventilation, and overall mortality [17,18]. Mortality in intensive care units due to COVID-19 is higher than that due to pneumonia of another origin [19]. Moreover, as the pandemic progressed, reported in-ICU mortality rates fell from more than 50% to just 40%, and the emergence of new strains can contribute to an increase in mortality rates [20]. Clearly, there is an urgent need for effective, safe, inexpensive, and compliant treatment during the pandemic. In this review we provided scientific data regarding the substantiation of the possibility of studying melatonin as a potential agent that might have some targets in COVID-19 pathogenesis. 

## 2. Melatonin—Nature and Origins

Melatonin plays an important role in physiologic processes and regulates body homeostasis [21]. It is synthesized from tryptophan in the pineal gland (epiphysis) and was identified in almost all organs of the body (like the liver, heart, skin, kidney, gut, placenta, etc.), providing a multifaceted universal influence on the reactivity parameters [21,22]. It has to be noted that 95% of all melatonin is secreted outside the pineal gland [23]. 

Chemically, melatonin is a small conservative indole molecule. Physiological effects of this compound are due to receptor-dependent and receptor-independent mechanisms of action [24]. Two melatonin membrane receptors in the central nervous system were cloned and pharmacologically characterized: MT1 (Mel1a) and MT2 (Mel1b) [25]. MT1 modulates neuronal firing, arterial vasoconstriction, cancer cell proliferation, and reproductive and metabolic functions; activation of MT2 modulates phase shift of circadian rhythms of neuronal excitation in the suprachiasmatic nucleus, suppression of dopamine release in the retina, induction of vasodilation and inhibition of leukocyte rolling in arterial channels, as well as the strengthening of immune responses [25]. It must be noted that melatonin-related receptors (MRR), nuclear receptors referred to as retinoid orphan receptors (ROR) or retinoid Z receptors (RZR), which include the RORα, RZRα, RORα2, and RZRβ, also have been identified to bind melatonin [24]. The receptor-independent mechanism of melatonin action will be reviewed in the following sections.

Thus, melatonin is a conservative molecule that has existed for billions of years (Figure 1) during which it has significantly perfected its functions. It is likely that such conservatism and long evolutionary existence can explain to some extent the variety of potential therapeutic effects.

## 3. “Bat-Coronavirus-Resistance” Hypothesis of Melatonin Benefits

In the previous review, A. Shneider, A. Kudriavtsev, and A. Vakhrusheva have hypothesized that the higher COVID-19 severity in the elderly is due to their reduced level of melatonin [26]. This hypothesis looks attractive because it is also supported by the fact that bats (natural carriers of coronaviruses) have almost no symptoms of the disease i.e., bats are naturally resistant to these viruses [27]. The melatonin level in bats ranges from 60 to 500 pg/mL at night, and 20–90 pg/mL during the day [28,29]. Human melatonin concentration is as follows (depends on age): 1–3 years old—260 pg/mL, 5–7 y.o.—160 pg/mL, 7–11 y.o.—100–110 pg/mL, 11–15 y.o.—80–85 pg/mL, 15–50 y.o.—50–55 pg/mL, 50–70 y.o.—27.8 pg/mL, and 70–90 y.o.—15.3 pg/mL [30]. Thus, there is a possibility that high melatonin levels in bats may contribute in some way to coronavirus resistance. In addition, a systematic review of 27 studies found that among humans, the risk of fatal COVID-19 is associated with age: the estimated age-specific infection fatality rate is very low for children and younger adults (e.g., 0.002% at age 10 and 0.01% at age 25) but increases progressively to 0.4% at age 55, 1.4% at age 65, 4.6% at age 75, and 15% at age 85 [31]. 

However, in addition to different melatonin concentrations, we believe there are other differences between the physiology of bats and humans. The differences in susceptibility to coronaviruses in bats and humans should be attributed to a significant difference in the mechanisms of the innate and adaptive immune response. Bats show several additional characteristics unique to mammals, such as a longer lifespan compared to body size, a low rate of carcinogenesis, and an ability to spread viruses without clinical manifestations of disease [32]. Moreover, it was demonstrated that SARS-CoV-2 receptor binding domain (RBD) binds to bat angiotensin-converting enzyme 2 (ACE2) with lower affinity than to human ACE2 [33]. On the other hand, it has been recently shown that melatonin administration stimulates the activity of dendritic reticular cells and macrophages through increasing the size, number, and endosomal compartments which may correlate to increased immunity; melatonin activates the proliferation and maturation of all immune cells including T and B lymphocytes, granulocytes, and monocytes [34]. Based on these data we can speculate that the high levels of this molecule can be beneficial for antiviral immunity. 

## 4. Aging, Oxidative Stress and Melatonin 

Clinical data show that at least 70% of ICU-hospitalized patients with COVID-19 have comorbid conditions. Patients with hypertension, cancer, immunodeficiency, obesity, and diabetes have a higher risk of severe COVID-19 infection or death [35]. For example, a study of 7300 patients with type 2 diabetes (T2D) showed three times higher mortality from COVID-19 compared to non-diabetic individuals [36]. Thus, comorbidities are a risk factor for a bad prognosis in patients with COVID-19 and it is known that with increasing age, the frequency of age-related diseases increases. 

One of the arguments in favor of melatonin use can be the revealed correlation between age and severity of COVID-19. Aging is an extremely complex and multifactorial, genetically and epigenetically mediated process, which is activated at the cellular level, including through oxidative stress and mitochondrial dysfunction [37]. Aging (with its comorbidities) further worsens SARS-CoV-2-caused increases of reactive oxygen (ROS) and nitrogen species (RNS) levels [38]. It can be assumed that COVID-19 enhances the constitutionally high age-dependent level of oxidative stress. 

Melatonin is known to have higher concentrations in mitochondria than in other organelles [39], but its level decreases significantly with age, which can trigger age-related changes [30]. In mitochondria melatonin provides antioxidant effects by removing radicals and reducing the degree of oxidative damage [39]—single melatonin molecule scavenges up to 10 ROS/RNS molecules [40]. Single electron transfer and hydrogen transfer are the major mechanisms by which melatonin ensnares radicals [41]. 

It is believed that after absorption, melatonin enters the mitochondria in large quantities—it is much more effective than synthetic antioxidants. Therefore, in addition to the fact that melatonin is produced in mitochondria, its additional exogenous intake ensures its high level in mitochondria [24]. In terms of the “oxidative stress theory of aging” it is expected that high levels of melatonin in mitochondria might protect the organism from the progression of age-related changes, and therefore will be useful during COVID-19 as well. In addition to the above, it should be noted that mechanical ventilation is also associated with evidence of early oxidative stress in the alveolar fluid and blood [42]. Since some patients with COVID-19 require mechanical ventilation, therapeutics are needed that can suppress oxidative stress. Thus, it has been proposed that melatonin may inhibit SARS-CoV-2-induced cell damage by regulating mitochondrial physiology [43]. 

## 5. ACE2, CD147 and Melatonin 

SARS-CoV-2 enters human cells using angiotensin-converting enzyme 2 (ACE2) which is highly expressed on type II alveolar epithelial cells, cholangiocytes, absorptive enterocytes from the ileum and colon, upper esophagus and stratified epithelial cells, cardiomyocytes, proximal tubule cells of the kidney, and bladder urothelial cells, etc. [44]. Comorbidities of cardiovascular disease, respiratory disease, diabetes, renal disease, and obesity are associated with higher levels of ACE2 [45]. There is indirect evidence that suggests a possible interference of melatonin in the SARS-CoV-2/ACE2 interaction, including through suppression of calmodulin, an essential intracellular component for maintaining ACE2 on the cell surface [46,47]. 

In addition to ACE2, another significant receptor for SARS-CoV-2 has also been identified—cluster of differentiation 147 (CD147) or basigin or extracellular matrix metalloproteinase inducer. CD147 is a transmembrane protein, that contributes to the development of tumors and bacterial and viral infections [48]. The study of RNA sequencing of human cells showed that ACE2 was expressed in lung and skin epithelial sites, while CD147 was expressed both in epithelial and immune cells. Asthma, COPD, hypertension, smoking, and obesity were associated with a higher expression of ACE2- and CD147-related genes in bronchial and blood cells. In addition, CD147-related genes were positively correlated with age and body mass index [49]. 

CD147 is a glycoprotein that is responsible for the cytokine storm manifestations in COVID-19 [50]. CD147 is involved in inflammation which develops through pro-inflammatory cytokines, such as interleukin-6 (IL-6), interferon-gamma (IFN-γ), tumor-necrosis factor-α (TNF-α), and monocyte chemo-attractant protein-1 (MCP-1) [51]. COVID-19 severity correlates with high levels of pro-inflammatory cytokines: TNF-α, IFN-γ, IL-6, IL-10, and C-reactive protein (CRP) [52,53]. Melatonin has previously been shown to reduce the remodeling effects of angiotensin II on the myocardium by blocking CD147 activity [54]. According to experimental studies, melatonin reduces the level of pro-inflammatory cytokines, such as IL-6 [55], and it can be assumed that this effect is also due to inhibition of the CD147-pathway. 

Thus, ACE2 and CD147 are key receptors for the entry of SARS-CoV-2 into human cells. These receptors play an important role in the progression of the disease and significantly affect the severity of its course. Therefore, blocking ACE2 and CD147 and/or their signaling pathways can have a significant effect on the course of COVID-19. As the studies we cited here show, melatonin can affect these receptors, which provides a pathophysiological basis for testing it in the fight against a pandemic. 

## 6. Inflammation and Melatonin

Cytokines secreted by macrophages in response to a viral load provoke a systemic inflammatory response with edema, acute respiratory distress syndrome, pneumonia, and multiple organ failure in elderly and comorbid patients. It is still far from clear why a cytokine storm is induced only in a fraction of patients with COVID-19 [56,57]. Cytokine storms are associated with high levels of tissue damage as evidenced by elevated plasma lactate dehydrogenase (LDH) and D-dimer levels (as pro-inflammatory cytokines) [58]. 

High LDH levels and leukopenia in severe COVID-19 indicate that leukocytes lose the integrity of the plasma membrane. It is believed that it is monocytes that must manage the balance between innate and adaptive immune responses, which can presumably be disrupted during a cytokine storm [59]. Leukopenia in severe COVID-19 patients appears to precede a cytokine storm [60]. It has also been shown that with other respiratory infections such as influenza A/H1N1, monocytes and macrophages are severely affected [61].

A recent study showed that SARS-CoV-2 recruits inflammasomes and induces pyroptosis in human monocytes (experimentally infected or collected from ICU-patients). Pyroptosis is mediated by activation of caspase-1, production of IL-1β, and increased levels of proinflammatory cytokines in primary human monocytes [59]. Previously, for SARS-CoV-1, it was demonstrated that a viral protein (encoded by the ORF8b gene) directly interacts with the NLRP3 inflammasome (nucleotide-binding domain leucine-rich repeat and pyrin domain containing receptor 3) [62]. NLRP3 via caspases leads to destruction of the cell membrane and the release of intracellular contents into the extracellular space [63]—it triggers pro-inflammatory IL-1β и IL-18 [64]—which predetermines the unbalanced release of IL-6 [59].

Experimental studies show that NLRP3 is an important link in immune inflammation and inhibition of this target can be an effective way to combat a cytokine storm [59]. In an LPS-induced acute lung injury mouse model, it was found that melatonin significantly reduced the pulmonary injury and decreased the infiltration of macrophages and neutrophils into the lung. It also was shown that NLRP3 inflammasome is activated by IL-1β and the caspase-1. Melatonin inhibits NLRP3 activation by suppressing extracellular histone release and directly blocking histone induced NLRP3 activation [65]. 

Another anti-inflammatory potential of melatonin is provided by the inhibition of nuclear factor-κappa beta (NF-κβ) and down-regulation of matrix metalloproteinases-3 (MMP-3), that modulates pro-fibrotic and pro-inflammatory cytokines [66]. This effect was shown in a rodent model of acute lung injury, where melatonin demonstrated protection of pulmonary tissue via inhibiting the activation of NF-κβ [67]. 

In a rat model of sodium nitrite-induced hypoxia, it was demonstrated that pretreatment with melatonin significantly reduced blood levels of extracellular heat shock protein 70 (Hsp70e), CRP, IL-6, and TNF-α [68]. A number of clinical studies, as well as a meta-analysis of 22 studies (with 749 participants) have shown that melatonin reduces the level of pro-inflammatory cytokines IL-6, TNF-α, and CRP [69,70,71,72,73]. Research data confirms the pronounced potential of melatonin in combating cytokine storms and hyperinflammation in patients with COVID-19. 

## 7. Fibrosis and Melatonin 

Fibrosis becomes the most dangerous complication after COVID-19 recovery. It depends on comorbidity, duration of ICU-stay, and mechanical ventilation [74]. Melatonin is known to prevent fibrosis through antioxidant and anti-inflammatory effects. In particular, it decreases oxidative stress after lung irradiation by enhancing the regulation of some enzymes, such as catalase, superoxide dismutase, glutathione, NADPH-oxidase 2 and 4, and by reducing the level of malondialdehyde [75]. Melatonin significantly reduced bleomycin-induced experimental lung fibrosis in mice and decreased transforming growth factor-β1-induced fibrogenesis in lung fibroblasts [76]. In patients with idiopathic cystic fibrosis, melatonin improved sleep and reduced nitrite in the exhaled breath condensate, which may also confirm the antifibrotic properties of this compound [77]. Another mechanism of antifibrotic action of melatonin was discovered by Lu Zhang et al. Authors demonstrated that melatonin reduced the production of ROS and prevented apoptosis and senescence in type II alveolar epithelial cells. Also, they found that melatonin significantly upregulated the expression of apelin 13 [78]. 

Therefore, taken together, all these data determine the need for research in the prevention and/or treatment of fibrosis in patients with COVID-19 using melatonin. 

## 8. Anxiety, Insomnia, and Melatonin 

The influence of the psychological factors on the outcomes of SARS-CoV-2 infection is widely discussed. On the one hand, general immunological reactivity is disrupted due to public anxiety as a result of attacks from the media, stress, and lack of sleep. On the other hand, the risk of psychological consequences of the COVID-19 pandemic, which manifest as anxiety, depression, insomnia, and, in some cases, suicide [79,80]. Moreover, impaired intellectual capacity, ability to abstract logical reasoning, planning and concentration, accompanied by haze, memory and attention problems, headache and depression, other signs of central nervous system damage, especially in people who have had severe COVID-19 and required the use of mechanical ventilation, can persist for many months [81]. 

Melatonin has powerful proven psychotropic effects and can reasonably be used in different categories of patients. It has anxiolytic, antidepressant, and sleeping effects, as well as the ability to reduce the consequences of stress [82,83]. The last is confirmed by an experimental study, where the combination of melatonin, vitamin C, and zinc was an effective preventive measure against severe psychological and chronic stress-induced biochemical manifestations of oxidative stress in rats due to abnormal conditions (e.g., SARS-CoV-2) [84]. According to a systematic review of 27 randomized controlled trials (involving 2319 participants), melatonin (doses varied from 3 to 10 mg per day) is more effective than a placebo in reducing anxiety. Moreover, it has a similar effect to benzodiazepines (midazolam and alprazolam), but melatonin is better tolerated [85]. Also, this compound significantly reduces the risk of depressive symptoms and anxiety in women with breast cancer [86], and in the same population melatonin improved sleep efficiency [87]. The meta-analysis of studies involving 1683 subjects showed that melatonin decreases sleep onset delay, increases total sleep time, and improves overall sleep quality [88]. Thus, using this safe over-the-counter drug, we can try to reduce the impact of psychological disorders and stress on patients with COVID-19 and after recovery, for “long-COVID”.

## 9. Clinical Data of Melatonin Administration in Patients with COVID-19

Using in silico systems biology-based drug repurposing screening method, it has been predicted that melatonin is a promising candidate to reduce COVID-19 progression and respiratory distress caused by a cytokine storm [89]. Of course, the in silico, in vitro, and animal studies we reviewed above cannot be easily extrapolated to humans. The large-scale use of melatonin for the treatment of COVID-19 requires controlled clinical trials. 

The first retrospective study (as a preprint) has been published on 791 patients with COVID-19 (948 intubation periods) and 2981 non-COVID-19 patients (3497 intubations) treated at New York Presbyterian/Columbia University Irving Medical Center. Melatonin exposure after intubation is significantly associated with a positive prognosis in COVID-19 and non-COVID-19 patients [90]. It might be explained by the positive effect of this compound on pulmonary pathology. 

Castillo et al. investigated the effect of 36–72 mg per day of oral melatonin among 7 cases with hospitalization due to COVID-19; none of the subjects died or needed mechanical ventilation [91]. An observational study of 26,779 persons from the COVID-19 registry showed that melatonin use is largely associated with a 28% decrease in the likelihood of a positive PCR-test for SARS-CoV-2 [92]. Interestingly, melatonin is more effective than angiotensin II receptor blockers or angiotensin-converting enzyme inhibitors and its preventive effect depends on age, gender, race, a history of smoking, and various comorbidities. 

A single-center, double-blind, randomized clinical study showed that melatonin at a dose of 3 mg three times per day for 14 days in hospitalized patients with confirmed mild to moderate COVID-19 significantly reduced clinical symptoms such as cough, dyspnea, and fatigue, as well as CRP levels and lung damage [93]. Similarly, at an oral dose of 9 mg per day for fourteen days, melatonin showed significant anti-inflammatory activity in hospitalized patients with COVID-19 by reducing and controlling the inflammatory cytokines IL-2, IL-4, and IFN-γ through regulation of Th1 and Th2 regulatory gene expression in patients [94]. The clinical observation showed a threefold reduction in the likelihood of delirium in patients hospitalized in ICU when melatonin was administered 3.5 mg/night (range: 1–10 mg) [95]. Another clinical study with 158 patients showed that the adjuvant use of melatonin (10 mg/night; in addition to standard therapeutic care) reduced the risk of thrombosis and sepsis, and decreased mortality in COVID-19 patients [96]. 

Nine clinical trials are currently underway on the effect of melatonin on COVID-19: NCT04474483 (40 mg/day for 14 days), NCT04784754 (9–900 mg/day for 14 days), NCT04409522 (9 mg/day for 7–10 days), NCT04531748 (100 mg/day for 3–14 days), NCT04568863 (5 mg/kg/day i.v., maximum 500 mg, for 7 days), NCT04530539 (10 mg/night for 14 days), NCT04353128 (2 mg prolonged release melatonin per day for 12 weeks for COVID-19 prevention), NCT04470297 (ramelteon—melatonin agonist 8 mg/night for 10 days), and NCT04570254 (50 mg/day up to 30 days) with total 1325 participants. 

In these trial protocols the daily dose of melatonin ranges from 2 to 900 mg. According to pharmacokinetic studies after 10 mg of oral melatonin administration mean *t*_max_ was 40.8 min with a median *C*_max_ of 3550.5 pg/mL; mean *t*_1/2elimination_ was 53.7 min; bioavailability 2.5%. Median *C*_max_ after i.v. bolus injection for the same dose of melatonin was 389,875.0 pg/mL; mean *t*_1/2elimination_ was 39.4 min [97]. Obviously, a dose of 10 mg will achieve a plasma concentration level higher than that of bats, however, fast melatonin excretion makes it necessary to take the drug continuously at regular time intervals or increase the total daily dose. Also, it is known that there is a significant difference between the physiological properties of melatonin and its therapeutic potential when used in non-physiological doses [98]. In human studies, doses of melatonin from 1 to 6.6 g per day for a duration of 30–45 days have shown no toxic effects [99]. It also was shown that 800 mg/kg was not lethal [100]. No side effects were reported in a clinical trial where 1400 women were administered with 75 mg of melatonin nightly for 4 years [101]. Finally, a meta-analysis of 50 studies, some of which were not blind, evaluated the efficacy of oral melatonin in doses ranging from 1 to 20 mg per day with a good safety profile [102]. 

Thus, melatonin should undoubtedly be extensively investigated for its therapeutic effect in the treatment of COVID-19. The wide dosage range and high safety profile make this drug even more attractive for further human studies. Russel J. Reiter and colleagues, by extrapolating the effective doses of melatonin used in animals to humans, suggested the use of melatonin as an adjunct to COVID-19 treatment in doses of 100–400 mg/day, especially if no efficient direct antiviral treatment is available [103]. We believe that the daily dose of melatonin in COVID-19 can even be increased to 1 g without any negative consequences (considering the safety of this dose and in the absence of other effective antiviral drugs) [99,100]. However, in some cases it could be more effective to divide the daily dose by 3–4 times in order to maintain a high concentration of the drug in the blood during the day. The latter can be supported by evidence that some patients may have decreased levels of melatonin, e.g., insulin resistance, glucose intolerance, and cardiovascular diseases are associated with low melatonin concentrations [103]. This conclusion is also justified by the excellent tolerability of the drug, as well as the short duration of administration—when there are symptoms of SARS-CoV-2 infection, i.e., on average about 14 days. 

In Figure 2 we schematically presented the most likely points of application of melatonin for the treatment of COVID-19. In Table 1 we summarized melatonin doses used in various animal and human studies.

## 10. Conclusions

In this review we have presented the most relevant evidence of the possibility of the successful use of melatonin to combat SARS-CoV-2 infection. Melatonin has a broad spectrum of potential pharmacological effects to contribute to COVID-19 treatment such as anti-inflammatory, anti-fibrotic, antiviral, antioxidant, and psychotropic. Given the wide range of possible dosing regimens, clinical trials are needed to find out the most effective doses and routes of administration. An important condition for the use of melatonin is availability as a medicinal product and nutritional supplement, as well as safety, and compliance. 

## Figures and Tables

**Figure 1 antioxidants-11-00140-f001:**
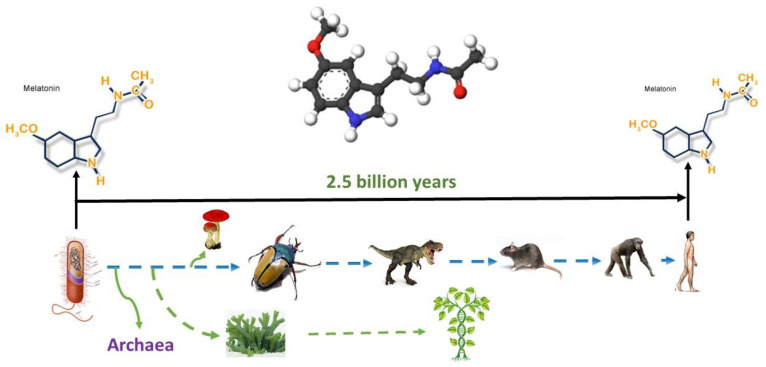
Melatonin originated 2.5 billion years ago, and it is present in all organisms from bacteria to humans; its structure has never been changed. The figure was obtained from the website http://www.melatonin-research.net (accessed on: 8 January 2022) with the permission of Dr. Dun-Xian Tan.

**Figure 2 antioxidants-11-00140-f002:**
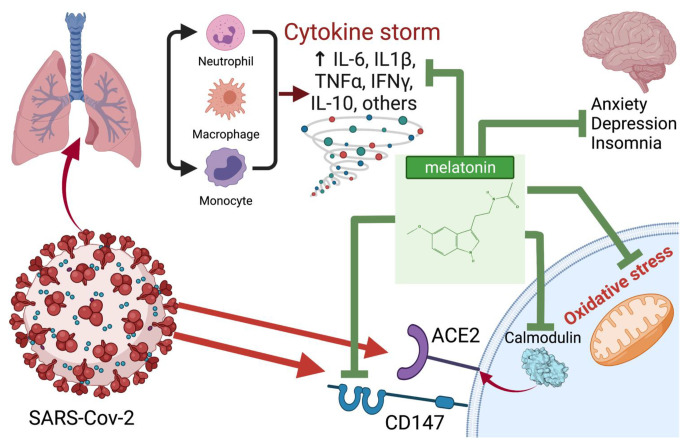
Potential points of application and probable mechanisms of the therapeutic effect of melatonin in COVID-19. Created with BioRender.com (accessed on: 8 January 2022).

**Table 1 antioxidants-11-00140-t001:** Doses of melatonin that have been used in various studies.

**Effects of Melatonin**	**Dose**	**References**
Animal studies		
Stimulates dendritic cells and macrophages	1 g (subcutaneous implant)	[34]
Decreases the levels of IL-6, TNF-α, leptin	1 mg/kg	[55]
Inhibits the activation of the NLRP3 inflammasome; reduces the level of IL-1β	30 mg/kg	[65]
Inhibits NF-κβ and down-regulates MMP-3	3 mg/kg	[66,67]
Reduces the levels of Hsp70e, CRP, IL-6 and TNF-α	200 mg/kg	[68]
Decreases oxidative stress after lung irradiation by enhancing the regulation of catalase, superoxide dismutase, glutathione, NADPH-oxidase 2 and 4, and by reducing the level of malondialdehyde	100–200 mg/kg	[75]
Reduces bleomycin-induced experimental lung fibrosis and decreases transforming growth factor-β1-induced fibrogenesis in lung fibroblasts	5 mg/kg	[76]
No lethality was observed	800 mg/kg	[100]
Human trials		
Reduces the levels of IL-6, TNF-α and CRP	3–25 mg/day	[69,70,71,72,73]
Improves sleep and reduces nitrite in the exhaled breath condensate (antifibrotic potential)	3 mg/day	[77]
Lowers sleep onset latency and increases total sleep time	3–6 mg/day	[82]
Decreases sleep onset delay, increases total sleep time, and improves overall sleep quality	0.5–5 mg/day	[88]
Reduces anxiety	3–10 mg/day	[85]
In women with breast cancer, reduces the risk of depression and anxiety, improves sleep	6 mg/day	[86,87]
Associated with survival of intubated COVID-19 patients	Not available	[90]
Adjuvant treatment in COVID19 pneumonia	36–72 mg/day	[91]
Associated with a 28% decrease in the likelihood of a positive PCR-test for SARS-CoV-2	Not available	[92]
Reduces COVID-19 symptoms, CRP levels, and lung damage	3 mg/day	[93]
Decreases the levels of IL-2, IL-4, and IFN-γ	9 mg/day	[94]
Reduces the risk of delirium in ICU-patients with COVID-19	1–10 mg/day	[95]
Reduces the risk of thrombosis and sepsis, decreases mortality in COVID-19 patients	10 mg/day	[96]
No side effects aside from drowsiness	1–6.6 g/day for 30–45 days	[99]
No side effects reported	75 mg/day for 4 years	[101]

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
