# Peer review of "Potential and Possible Therapeutic Effects of Melatonin on SARS-CoV-2 Infection"

_antioxidants, 2022, doi:10.3390/antiox11010140_

Round 1
Reviewer 1 Report
The paper described melatonin's effects on several diseases and its implication on protecting SARS-CoV-2 infection. The review was well written, and melatonin effects are well organized.
One possible comment is that it is easy to understand the authors could insert the Table summarizing the ED50 or effective dose of each melatonin-related protective effect on each disease.
Author Response
Thanks a lot for the recommendation. We have revised manuscript according to your comments.
Point 1. One possible comment is that it is easy to understand the authors could insert the Table summarizing the ED50 or effective dose of each melatonin-related protective effect on each disease.
Response 1. We added the Table 1 in the main text.
Reviewer 2 Report
This manuscript entitled “Potential and Possible Therapeutic Effects of Melatonin on SARS-CoV-2 Infection” reinforces the protective role of melatonin in COVID-19 disease. The paper is well written, is concise, and shows results with potential therapeutic interest of melatonin, arranged in prominent sections.
Minor request:
-Line 237: leave one space under the title 8.
-Line 255: “melatonin (doses varied from 3 to 10 mg) is more effective than placebo in 255 reducing anxiety”- specify if the dose is per day.
-Line 321: “Russel J. Reiter and colleagues (…) suggested to use melatonin as an adjunct to COVID-19 treatment in doses of 100-400 mg/day”- it should be indicated too the following text: “especially if no efficient direct anti-viral treatment is available”, such as the cited sentence reference.
-Line 324: the sentence “We believe that the daily dose of melatonin in COVID-19 can be even increased to 1 g without any negative consequences” should be accompanied by some reference to reinforce such a claim.
-Line 325: authors should consider that such as Reiter et al (2020) suggest “the daily dose of ∼3 mg to a maximum of 10 mg, 30–60 min before bedtime to better simulate the normal physiological circadian rhythm of melatonin”, for preventing any problem, e.g., delirium, in those patients that may have a disturbance of wake-sleep rhythm. When they affirm “in order to maintain a high concentration of the drug in the blood during the day” perhaps it would be necessary to consider this way of administrations, during the day, in those cases where there are decreased blood melatonin levels, such in patients with insulin resistance or glucose intolerance, what have been previously documented.
-Line 335-338: repeats the same text as in line 12-15 (abstract): “We have attempted to present scientifically proven mechanisms of action for the potential therapeutic use of melatonin during SARS-CoV-2 infection. A wide range of pharmacological 13 properties allows to include it as an effective addition to the methods of prevention and treatment of COVID-19”. Whereas in conclusions it should be necessary to highlight the most important things in relation to melatonin and COVID, specifying key points.
Author Response
Thanks a lot for the review and helpful recommendations. We have fixed everything according to your advice.
Point 1. -Line 237: leave one space under the title 8.
Response 1. Done.
Point 2. -Line 255: “melatonin (doses varied from 3 to 10 mg) is more effective than placebo in 255 reducing anxiety”- specify if the dose is per day.
Response 2. Added "per day"
Point 3. -Line 321: “Russel J. Reiter and colleagues (…) suggested to use melatonin as an adjunct to COVID-19 treatment in doses of 100-400 mg/day”- it should be indicated too the following text: “especially if no efficient direct anti-viral treatment is available”, such as the cited sentence reference.
Response 3. Added the following text: “especially if no efficient direct anti-viral treatment is available”
Point 4. -Line 324: the sentence “We believe that the daily dose of melatonin in COVID-19 can be even increased to 1 g without any negative consequences” should be accompanied by some reference to reinforce such a claim.
Response 4. We've added the reference and clarification that this judgment is based on the safety of this dose and only in the absence of other alternative therapies.
Point 5. -Line 325: authors should consider that such as Reiter et al (2020) suggest “the daily dose of ∼3 mg to a maximum of 10 mg, 30–60 min before bedtime to better simulate the normal physiological circadian rhythm of melatonin”, for preventing any problem, e.g., delirium, in those patients that may have a disturbance of wake-sleep rhythm. When they affirm “in order to maintain a high concentration of the drug in the blood during the day” perhaps it would be necessary to consider this way of administrations, during the day, in those cases where there are decreased blood melatonin levels, such in patients with insulin resistance or glucose intolerance, what have been previously documented.
Response 5. We carried a clarification on your recommendation (lines 325-329).
Point 6. -Line 335-338: repeats the same text as in line 12-15 (abstract): “We have attempted to present scientifically proven mechanisms of action for the potential therapeutic use of melatonin during SARS-CoV-2 infection. A wide range of pharmacological 13 properties allows to include it as an effective addition to the methods of prevention and treatment of COVID-19”. Whereas in conclusions it should be necessary to highlight the most important things in relation to melatonin and COVID, specifying key points.
Response 6. We have changed this section. We made a conclusion on the key positions on which melatonin may be a potential treatment for COVID.
Reviewer 3 Report
the review by Shchnetinin et al. reviews the current knowledge on the potential role of melatonin during SARS-CoV-2 infection. The manuscript is quite short; however, I believe that it is a huge advantage of it. It perfectly summarizes the current knowledge. There are no unnecessary long introductions, the references are correct, the graphical summary is perfect, proper data is described. I believe that the manuscript deserves to be published the way it is. Good job!
Author Response
We are very grateful to you for such a positive response! Thanks a lot! Our entire team are thrilled with such a positive review!
Best wishes,
Dr. Albert Bolatchiev
This manuscript is a resubmission of an earlier submission. The following is a list of the peer review reports and author responses from that submission.